# Increasing Nebulizer Spray Efficiency Using a Baffle with a Conical Surface: A Computational Fluid Dynamics Analysis

**DOI:** 10.3390/bioengineering12070680

**Published:** 2025-06-20

**Authors:** Hung-Chieh Wu, Fu-Lun Chen, Yuan-Ching Chiang, Yi-June Lo, Chun-Li Lin, Wei-Jen Chang, Haw-Ming Huang

**Affiliations:** 1School of Dentistry, Taipei Medical University, Taipei 11031, Taiwan; m204111012@tmu.edu.tw (H.-C.W.); m8404006@tmu.edu.tw (W.-J.C.); 2Division of Infectious Diseases, Department of Internal Medicine, Wan Fang Hospital, Taipei Medical University, Taipei 11696, Taiwan; 96003@w.tmu.edu.tw; 3Department of Mechanical Engineering, Chinese Culture University, Taipei 11114, Taiwan; jyj3@ulive.pccu.edu.tw; 4Division of Periodontics, Department of Dentistry, Wan Fang Hospital, Taipei Medical University, Taipei 11696, Taiwan; lopa@w.tmu.edu.tw; 5Department of Biomedical Engineering, Medical Device Innovation and Translation Center, National Yang Ming Chiao Tung University, Taipei 112304, Taiwan; cllin2@nycu.edu.tw

**Keywords:** breath-actuated nebulizer, CFD, baffle

## Abstract

Breath-actuated nebulizers used in aerosol therapy are vital to children and patients with disabilities and stand out for their ability to accurat ely deliver medication while minimizing waste. Their performance can be measured according to the mass output and droplet size. This study aimed to analyze how the baffle impact surface geometries affect the pressure and flow streamlines inside the nebulizer using computational fluid dynamics (CFD). Computer-aided design models of conical symmetric, conical asymmetric, and arc-shaped baffle designs were analyzed using CFD simulations, with the optimal spray output validated through the differences in mass. Conical baffles exhibited superior pressure distribution and output streamlines at 0.25 cm protrusion, suggesting that the nebulizer spray performance can be enhanced by using such a conical baffle impact surface. This result serves as a valuable reference for future research.

## 1. Introduction

The lungs are responsible for respiratory functions that deliver oxygen to the blood and expel carbon dioxide from the body, playing a crucial role in the body’s gas exchange process [1]. However, they are also susceptible to diseases that can have a severe impact on both quality of life and lifespan. Common lung diseases include chronic obstructive pulmonary disease (COPD), pneumonia, asthma, and pulmonary fibrosis, all of which cause damage to the lungs. The most effective current treatment for these lung diseases is aerosol therapy [2].

The lungs have a large surface area, thin alveolar–blood barrier, low enzyme activity, and reduced first-pass metabolism, which give lung-delivered nebulized drugs high bioavailability [3]. Aerosol therapy is commonly used to deliver medications in mist form for treating patients with lung diseases [4], and it is particularly effective in addressing the challenges of administering medication to patients with respiratory system disorders or symptoms [5]. This therapy works by converting drugs or drug solutions into inhalable fine mist particles that deliver a high concentration of medication directly into lung tissues [6]. Aerosol therapy used in respiratory disease treatment has been widely researched across conditions including acute respiratory distress syndrome (ARDS) [7], chronic obstructive pulmonary disease [8], cystic fibrosis [9], ventilator-associated pneumonia [10], dyspnea [11], and acute asthma [12]. 

The primary function of a nebulizer in aerosol therapy is to convert medication into fine aerosols or mist that patients can inhale into their lungs through breathing [13]. Such devices allow medications to act directly on respiratory tissues in order to treat respiratory diseases and alleviate symptoms [14]. There are various types of nebulizers available on the market, such as jet nebulizers [15], traditional ultrasonic nebulizers, vibrating mesh nebulizers, soft mist inhalers [16], air pump nebulizers, mesh nebulizers, metered-dose inhaler nebulizers, clinical nebulizers, oral–nasal mask nebulizers, and breath-actuated nebulizers [17]. Breath-actuated nebulizers generate aerosol only upon inhalation, a feature that provides precise drug dosing [18]. Additionally, breath-actuated nebulizers address the drawbacks of traditional continuous-output nebulizers, which release medication during both inhalation and exhalation, resulting in both wasted medication and environmental contamination. Previous studies have shown that breath-actuated nebulizers can increase the inhaled aerosol by three to four times compared to traditional continuous-output nebulizers [18]. They do not require manual operation, which reduces caregiving workloads in isolated intensive care units. However, breath-actuated nebulizers have limitations such as lower medication output and longer treatment times in pediatric patients aged 2 to 4 years with insufficient respiratory effort [19]. Improving the output efficiency of breath-actuated nebulizers has thus become an important topic of research.

Computational fluid dynamics (CFD) is a numerical simulation technique that uses mathematical equations and computational methods to approximate complex fluid dynamics phenomena. It is employed to simulate and analyze fluid behavior in various applications, including the flow of gases and liquids, heat transfer, mass transfer, and their interactions [20]. CFD can simulate real-world fluid behaviors in settings such as wind tunnel testing [21], automotive design [22], weather forecasting [23], and environmental prediction [24]. CFD analytical methods include the finite volume method (FVM) [25], the finite element method (FEM) [25], and the finite difference method (FDM) [26]. Most CFD analyses related to nebulizers focus on the fluid dynamics of aerosolized drugs in the respiratory airway [27,28,29,30]. Although numerical methods were reported as an effective tool for studying inhaler design [31], research on the structural design of nebulizer devices is limited. In this study, FVM-based CFD was used to analyze the breath-actuated nebulizer and explore potential improvements to achieve the maximum spray output with minimal breath force.

## 2. Materials and Methods

### 2.1. Breath-Actuated Nebulizer Operation

The breath-actuated nebulizer used in this study (AllNeb, Enchant Tek Co., Ltd., Yilan, Taiwan) is an external mixing nebulizer that operates in two phases [13]. When the patient inhales, the suction force in the oral cavity generates negative pressure, which opens an air intake valve at the top of the nebulizer. External air then enters the nebulizer and equalizes the internal and external pressure. This causes a baffle to move downward, allowing high-speed airflow to drive the medication solution upward to impact the baffle and aerosolizing the medication. The negative pressure created by the patient’s inhalation causes the aerosol particles inside the nebulizer to move outward, thus delivering the drug. Upon exhalation, the pressure causes an air intake valve at the top of the nebulizer to close, creating internal pressure that drives the baffle upward to stop both aerosolization and drug output.

### 2.2. CFD Modeling

Computer-aided design software (SolidWorks 2020, Dassault Systèmes S.A., Vélizy-Villacoublay, France) was used to create a solid model of the breath-actuated nebulizer flow channel (Figure 1a). The model contained all the internal parts of the nebulizer (Figure 1b,c). As the nebulizer’s external shell was not an object of this study, the nebulizer was enclosed by a solid external cuboid that covered all the flow channels to become the boundary of the flow channel of the model. The surfaces of the upper inflow port, drug spout port, and compressed air inlet were located on the outer surface of the external solid cuboid (Figure 1d).

Commercial CFD software was used to perform the calculations (ANSYS Fluent 2020 R2, Swanson Analysis Systems, Inc., Canonsburg, PA, USA). The geometric model of the nebulizer (Figure 1) was stored as a step file, imported into ANSYS Fluent, and meshed using the ANSYS auto-mesh function. Air was set as the flow channel material, while the external cuboid material was set as a solid (Figure 2a and Figure 2b, respectively). The model comprised 299,945 elements and 62,534 nodes. The convergence criterion was met when changing the element size three consecutive times no longer affected the computational results [32]. Such an element size setting met the accuracy requirements stated in the literature [33]. For the boundary condition, the degrees of freedom in the X, Y, and Z directions for the nodes on the boundary wall surface were set to zero.

To calculate the flow fields inside the nebulizer, the upper inflow port and compressed air inlet flow rates were set as 5 and 10 LPM (L/min), respectively (Figure 2c,d). The drug spout port flow was set as the flow inside the nebulizer with a flow rate of −15 LPM. According to the ANSYS requirements, the flow rates were converted into the fluid velocity (upper inflow port 0.12 m/s, compressed air inlet 1.25 m/s, output port 0.031 m/s). A total of 200 internal flow velocity lines were calculated. Fifty flow velocity lines were plotted during the post-process in ANSYS for ease of observation. The flow velocity lines and pressure contours were calculated and compared to demonstrate the performance of nebulizers with different baffle designs.

### 2.3. Parameter Settings of the CFD Model

This study entailed an analysis of the impact of different surface baffle shapes on the airflow inside a nebulizer, and it included a flat baffle (Figure 3b), a conical symmetric baffle (Figure 3b), an arc-shaped baffle (center distance 1 mm, rounded corner radius 2 mm, Figure 3d), and a conical asymmetric baffle (Figure 3e). Among these, the conical symmetric baffle exhibited the best flow. To further analyze the optimal flow characteristics, the height of the symmetrical conical baffle was modeled at 0.15 mm, 0.25 mm, and 0.35 mm (Con-0.15, Con-0.25, Con-0.35). The resulting changes in the flow lines were compared with those of the flat and arc-shaped baffles (Flat, Arc-0.25).

### 2.4. Nebulizer Efficiency Measurement

To validate the CFD results, the physical spray distance and spray mass were measured with 3D-printed baffle components installed in the nebulizers (Figure 4a–e). A camera was used to capture a one-minute video recording of the maximum spray distance (Figure 5). The spray was created using pressurized air (compressor, 9R-021000, Wellell Inc., Taipei, Taiwan). The mass was measured using absorbent papers fixed at the spray outlet. Both the absorbent paper and the nebulizer were dried overnight in an oven and weighed before the experiment. During the experiment, the absorbent paper was fixed 0.3 cm ± 0.1 cm from the nebulizer outlet. In this experiment, a flat baffle was used as a control design. Six milliliters of deionized water were placed in the nebulizer with a flat baffle in the nebulizer, and spraying was conducted for 5 min. The absorbent paper and nebulizer were then weighed separately on a microbalance, with the differences in mass before and after the experiment used for the statistical analysis and between-groups comparison. All the experiments were repeated five times. The results are presented as the mean ± standard deviation. The differences between groups were analyzed using one-way ANOVA, and a Scheffe post hoc test was performed to demonstrate the difference. Statistical significance was defined as *p*-values lower than 0.05 for all the analyses.

## 3. Results and Discussion

Pneumatic nebulizers are aerosol-generating devices used to convert liquids into aerosols of a size that can be inhaled into the lower respiratory tract [13]. Previous studies have shown that breath-driven nebulizers can effectively reduce lung hyperinflation and the respiratory rate in patients with chronic obstructive pulmonary disease (COPD) [34]. Aerosol drug delivery follows the principles of fluid dynamics, making CFD, which is typically used in physics simulations, a suitable method for analyzing nebulizers [31]. The CFD analysis shows simulated streamlines in the breath-actuated nebulizer model (Figure 6a). When liquid medication is aerosolized using compressed air, it mixes with the stream and flows upward (Figure 6c), striking the baffle positioned in the aerosol path. Upon impact, the stream disperses in all directions (Figure 6c–e), as reported previously [35]. These high velocity streams create a low-pressure area near the nozzle (Figure 7 and Figure 8), consistent with a previous study [13]. These results can be explained by Bernoulli’s principle, which predicts that as the velocity of a fluid increases, its pressure decreases. While the nebulizer chamber, with the exception of the compressed air outlet, is under positive pressure, the nebulizer outlet is under low negative pressure (Figure 7). This pressure distribution causes the airflow inside the nebulizer to flow from the chamber toward the outlet, confirming that the FE model used in this study can accurately simulate actual breath-actuated nebulizer spray operation. 

When the nebulizer is operating, if the airflow path inside the nebulizer is longer, the loss of kinetic energy during flow will be greater. In addition, if the airflow passes through the surface of liquid medication, it will not be able to carry the aerosolized medication out of the nebulizer. These conditions will all reduce the efficiency of the nebulizer in terms of delivering aerosolized medication. In addition, when the airflow passes over the inner surface of the nebulizer, large droplets were observed adhering to the nebulizer walls, which also led to reductions in the amount of medication delivered [13]. When using a conventional flat baffle, the flow lines within the nebulizer pass through the nebulizer’s lower part and pass through the liquid medication surface, then approach the inner wall of the nebulizer (Figure 6a,b). This may be because the jet flow from the nozzle forms a stream wall [13] after hitting the baffle, thereby blocking the path for the airflow to pass through the center (Figure 6c–e). When the bottom of the baffle is designed as an asymmetric cone, symmetric cone, or arc, the stream film observed in the flat baffle model does not form (Figure 9c,f,i,l). The flow lines instead pass through the central area (Figure 9e,h,k), away from the medication surface and inner wall. This is because of the upward flow of the compressed air jet from the nozzle at high speed after striking the baffle (Figure 9f,i,l). This creates an upward negative pressure gradient (Figure 10), which attracts the incoming air toward this central position. This helps the aerosolized medication mix with the airflow and be carried out from the air inlet. However, the air inlet flow streams near the nozzle ports with asymmetric conical and arc-shaped (Figure 9e,h) baffles show significant vortices, making observation of the airflow exiting the nebulizer difficult. Although the conical baffle design causes vortex formation at the leading edge of the baffle, due to the compressed air jet flow, airflow can still be observed exiting the nebulizer (Figure 9i). A nebulizer design that incorporates a symmetric conical baffle appears to be optimal for efficient spray.

This study also includes simulations to identify the optimal conical baffle size. The Con-0.25 nebulizer outlet position had the most flow lines, indicating the best spray efficiency (Figure 10a). This is due to the Con-0.25 nebulizer having the largest negative pressure area near the nozzle and maintaining a pressure gradient from bottom to top throughout the entire nebulizer (Figure 10b). In addition, the positive outlet pressure in the Con-0.15 and Con-0.35 nebulizers was equal to or greater than the internal pressure of the nebulizer (Figure 10b) and became the reason why the flow lines did not enter the spray outlet. 

Several factors can influence the performance of a nebulizer, including the nebulization time, cost, and ease of use [13]. While the output tends to increase with higher flow rates used to power the nebulizer [36,37], the optimal driving flow is limited to a specific range with a flow rate of 8–10 LPM recommended [36]. Flow rates below this level can reduce the nebulizer efficiency, while higher flow rates may lead to increased drug loss during the exhalation phase. The nebulizer’s driving flow in the current CFD analysis was set at 10 LPM, aligning closely with the recommended value. However, the flow line, pressure contour, and output performance vary depending on the baffle design used. When five different nebulizer baffle designs were compared, the Con-0.25 baffle had the greatest aerosol output distance (32.5 cm), approximately 41% greater than that of the flat baffle (23 cm) (Figure 11). Although the Arc-0.25 baffle also had better spray performance (27.5 cm) compared to the flat baffle, the improvement was only 15.5%, while the nebulizer with the Con-0.15 baffle had a shorter spray distance (21 cm) than the conventional flat baffle. These results are consistent with the CFD analysis, both indicating superior spray efficiency with the Con-0.25 baffle design. 

Using the traditional mass loss method to measure the nebulizer output has certain drawbacks. The nebulizer output includes both vapor and aerosol components, with the proportion of vapor being influenced by the liquid temperature in the reservoir [38]. Estimations of the aerosol output based on the mass difference before and after nebulization overlook the effects of evaporation, resulting in aerosol output losses ranging from 15% to 54% and experimental outcomes that overestimate the actual aerosol output [38]. Since this study aimed to compare the effects of different baffle designs on the nebulizer spray volume, the actual spray volume was not the primary focus of this research.

To assess the experimental setup, three absorbent papers were randomly selected to conduct spray absorption experiments. Performing mass loss experiments with different absorbent papers did not result in statistically significant differences. The spray mass of the Con-0.15 (0.44 ± 0.07 g) and Con-0.35 baffles (0.60 ± 0.04 g) showed no significant differences compared to the flat baffle (0.55 ± 0.05 g). However, for the Arc-0.25 and Con-0.25 baffles, the nebulizer’s spray masses were measured at 0.65 ± 0.07 g and 0.80 ± 0.01 g, respectively, both significantly greater than that of the flat baffle (*p* < 0.01). Additionally, the spray mass when using the Con-0.25 baffle was significantly greater than that of the Arc-0.25 baffle (*p* < 0.01) (Figure 12b). These results are consistent with the CFD analysis, confirming that the conical baffle has better spray performance than the traditional flat-designed baffle and that the Con-0.25 baffle design is more efficient than the other baffle shapes.

During the operation of the nebulizer, the aerosolized medication is carried out through the drug spout port by the airflow from the upper inflow port. As shown in Figure 10b, the Con-0.25 nebulizer generates a larger negative pressure region around the tip of the baffle. This negative pressure draws in air from the upper inflow port, which in turn helps carry a greater amount of aerosolized medication out of the nebulizer. The degree of medication atomization is related to the distance between the compressed air outlet and the baffle tip. If the distance is too short, the aerosolized medication may move in the opposite direction, obstructing the upward flow path of compressed air. Conversely, if the distance is too great, excessive kinetic energy is lost before reaching the baffle, resulting in insufficient impact force. Therefore, optimal atomization occurs at a specific compressed air outlet port-to-baffle distance. In this study, we found that a distance of 0.25 mm provides the best design.

Although this study successfully identified improvements to the baffle design and nebulizer spray performance, several limitations could be addressed in future studies. Although the droplet size is an important factor relating to nebulizer performance [13,39], several factors influencing the aerosol droplet size, including the solution temperature [40], viscosity [41], and gas velocity [42], were not included in this study, leaving room for further optimization of the nebulizer design. To address this issue, the introduction of two-phase (solid–liquid mixture) CFD analysis will be a direction for future research. In addition, the introduction of medication alters the internal space of the nebulizer, which must affect the airflow path. Since the CFD model in this study does not simulate the presence of the liquid medication, it becomes another limitation of this study.

## 4. Conclusions

For the methodology, breath-actuated nebulizers are generally not recommended for children under the age of five, as their breathing effort is often insufficient to activate the trigger valve. Although this study does not directly analyze the actuation of the trigger valve in breath-actuated nebulizers, the CFD-based analysis presented in this study may serve as a reference for structural designs aimed at reducing the activation force. This could help address the issue of breath-actuated nebulizers being unsuitable for younger children. In terms of the results, this study shows that the aerosol drug output efficiency of the breath-actuated nebulizer is related to the shape of the baffle tip and its distance from the compressed air outlet. Our results demonstrated that conical baffles exhibited superior pressure distribution and output streamlines at 0.25 cm protrusion, suggesting that the nebulizer spray performance can be enhanced by using such a conical baffle impact surface.

## Figures and Tables

**Figure 1 bioengineering-12-00680-f001:**
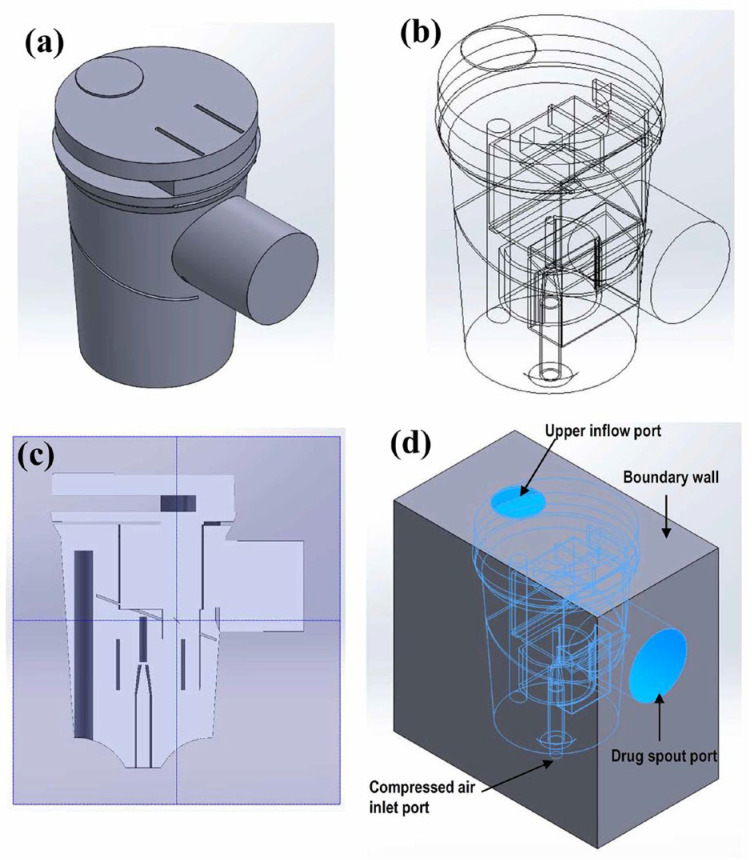
Geometric model of the nebulizer flow chamber. (**a**) CAD software model. (**b**) Internal flow channel structure. (**c**) Model cross-section. (**d**) Simulated outer cuboid boundary.

**Figure 2 bioengineering-12-00680-f002:**
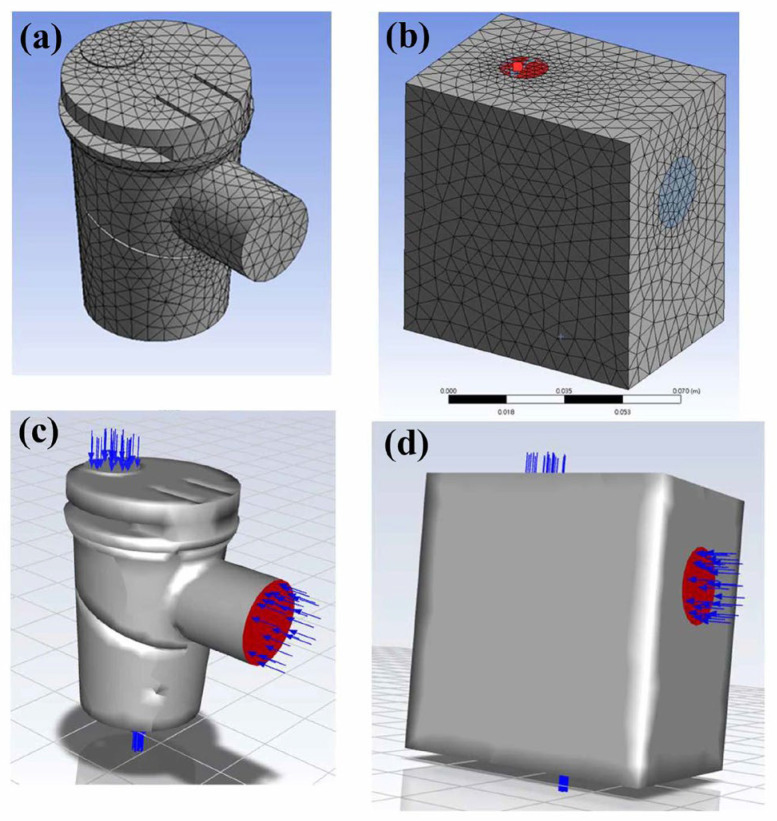
CFD model of the nebulizer flow. (**a**) Model meshed using tetrahedral elements. (**b**) Meshed outer cubic wall. (**c**) The outlet flow rate is set as the negative value of the flow into the nebulizer. (**d**) The entire model with the boundary conditions.

**Figure 3 bioengineering-12-00680-f003:**
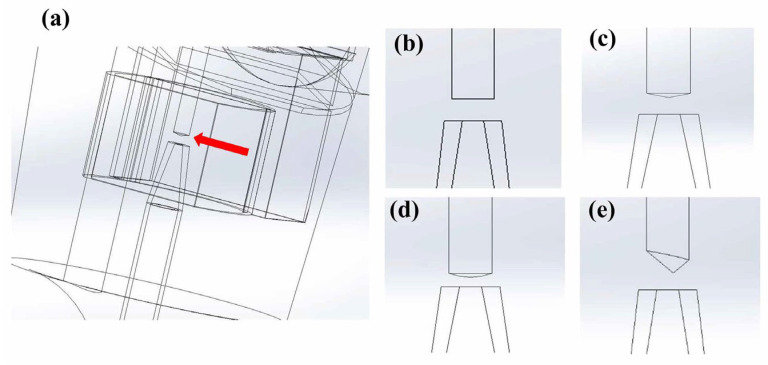
(**a**) Baffle location within the nebulizer. The baffles exhibit (**b**) traditional flat, (**c**) conical symmetric, (**d**) arc, and (**e**) conical asymmetric impact surface geometries. The red arrow identifies the location of the baffle tip.

**Figure 4 bioengineering-12-00680-f004:**
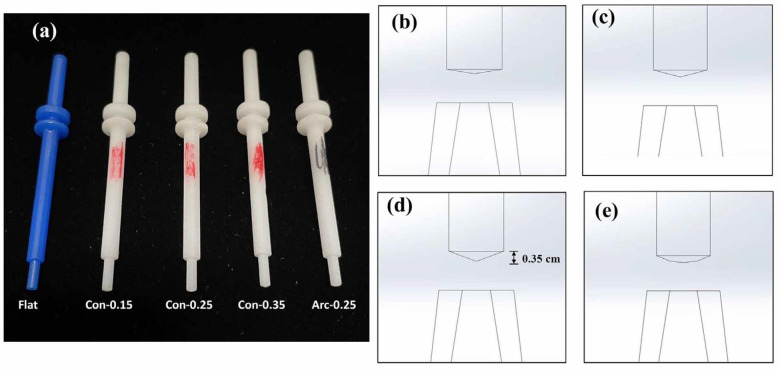
(**a**) Three-dimensional-printed baffles used for mass difference experiments. Conical symmetric baffles with protrusions of (**b**) 0.15 cm, (**c**) 0.25 cm, and (**d**) 0.35 cm. (**e**) Arc surface design.

**Figure 5 bioengineering-12-00680-f005:**
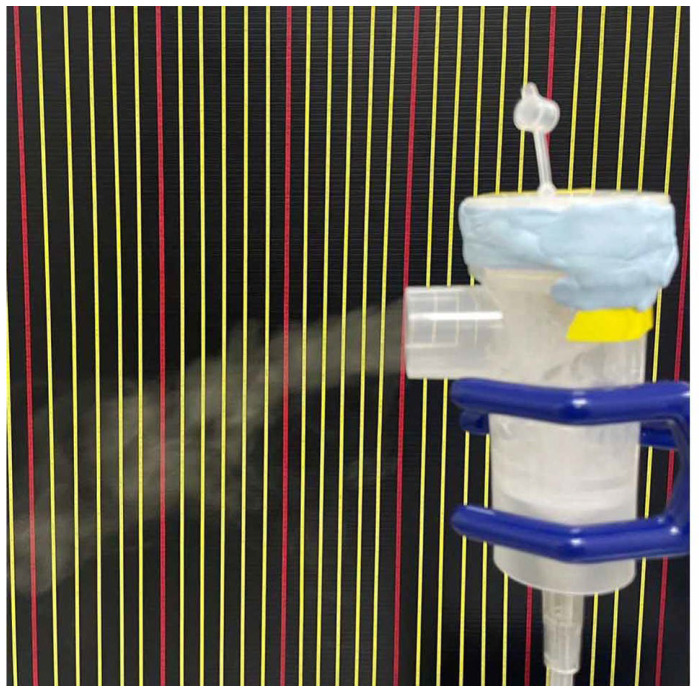
Experimental setup of the spray distance. The vertical line on the background provided reference markers (6 cm between red lines). The data measured the distance from the start to the final points of the background markers.

**Figure 6 bioengineering-12-00680-f006:**
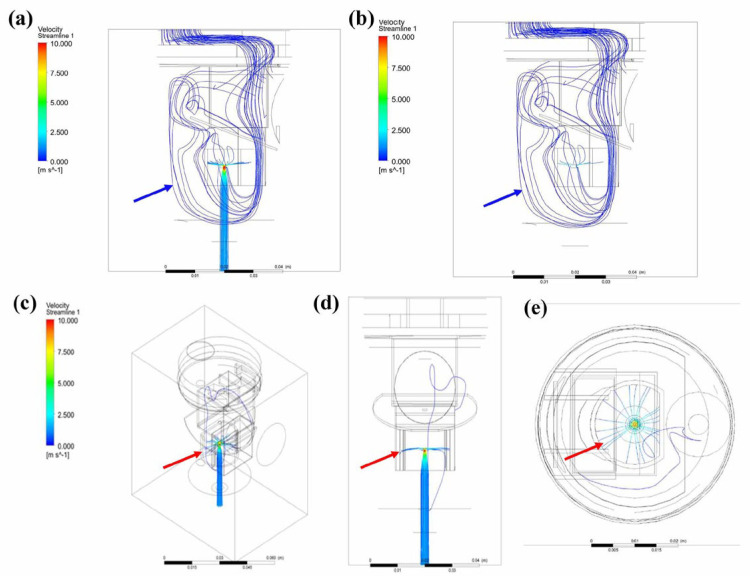
Flow lines in the simulated nebulizer with a traditional flat baffle. (**a**) Upper air inlet port and lower compressed air input, side view. (**b**) Flow lines within the nebulizer pass through the lower medication part and approach the inner wall of the nebulizer (blue arrows). (**c**) Three-dimensional view, (**d**) sagittal view, and (**e**) top view of the stream film created by the jet flow hitting the baffle (red arrows).

**Figure 7 bioengineering-12-00680-f007:**
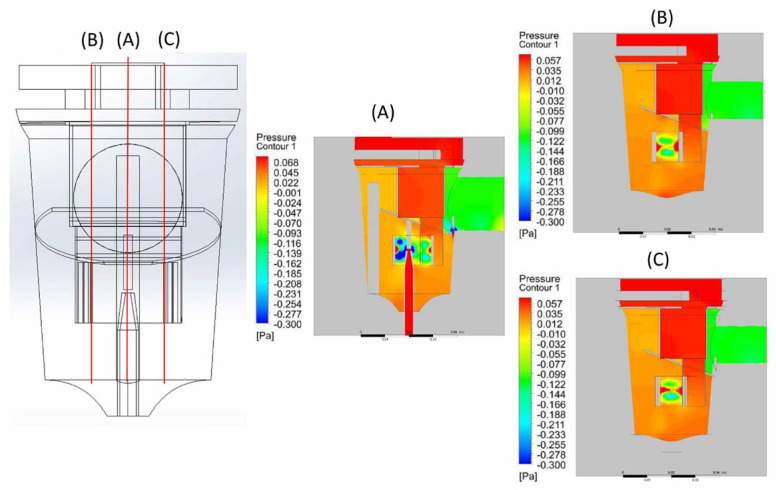
Side view of the pressure contour. (**A**) Pressure distribution at the sagittal plane. (**B**,**C**) Pressure patterns at the parasagittal planes.

**Figure 8 bioengineering-12-00680-f008:**
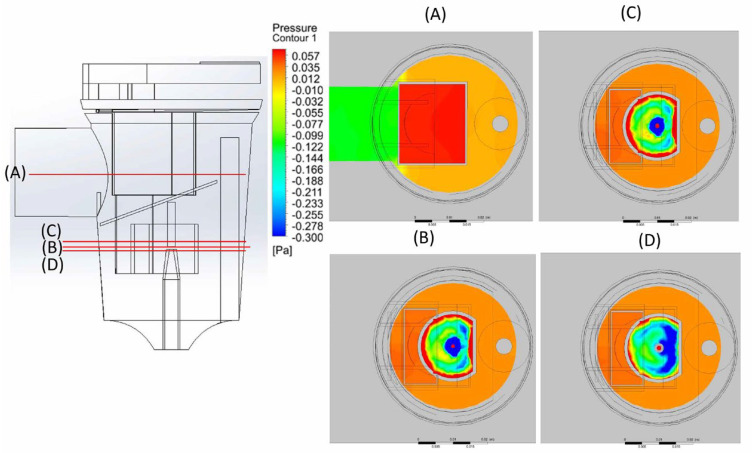
Top view of the pressure contour. (**A**) Pressure along the outlet port central plane. (**B**) Pressure patterns at the nozzle plane and the surrounding areas (**C**,**D**).

**Figure 9 bioengineering-12-00680-f009:**
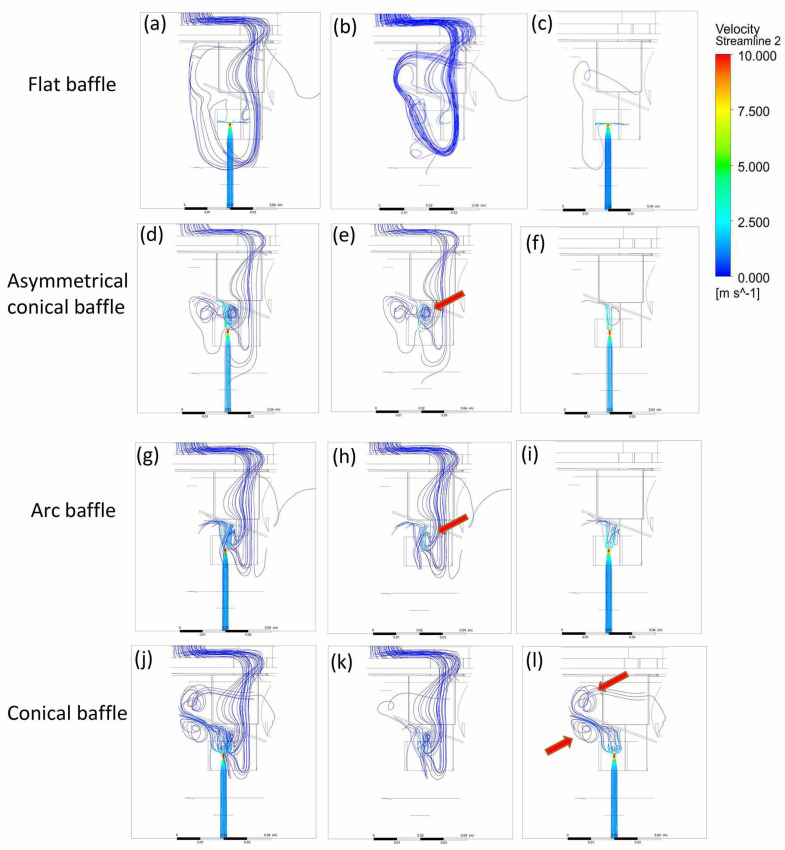
Flow lines with (**a**–**c**) flat, (**d**–**f**) conical asymmetric, (**g**–**i**) arc, and (**j**–**l**) conical baffles. Red arrows indicate vortex locations.

**Figure 10 bioengineering-12-00680-f010:**
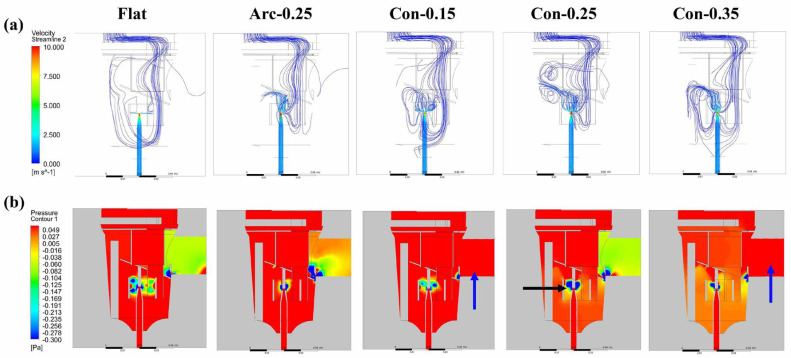
(**a**) Streamline and (**b**) pressure contours due to the nozzle jet flow. The Con-0.25 baffle exhibited the most (**a**) output flow lines and (**b**) the largest negative pressure area near the nozzle (black arrow). The Con-0.15 and Con-0.35 outlet pressures were positive and equal to or greater than the nebulizer internal pressure (blue arrows).

**Figure 11 bioengineering-12-00680-f011:**
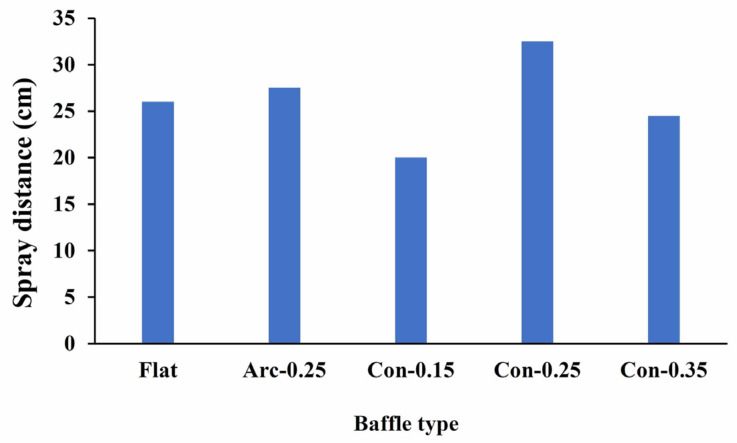
The Con-0.25 baffle had the greatest aerosol output distance, approximately 41% greater than the output distance of the flat baffle.

**Figure 12 bioengineering-12-00680-f012:**
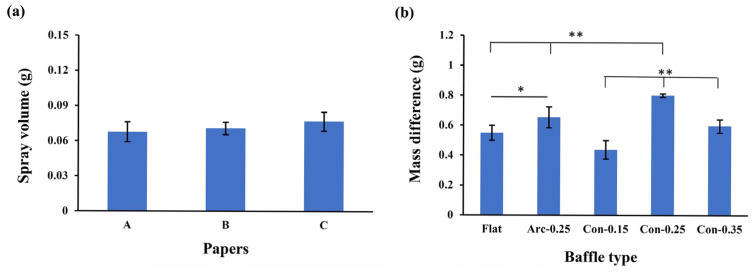
(**a**) No statistical difference in the mass change was found between the absorbent papers at the spray outlet when a flat baffle was used in the nebulizer. (**b**) The Con-0.25 baffle exhibited the greatest spray mass compared to the other baffle designs. * and ** denote *p* values lower than 0.05 and 0.01.

## Data Availability

The original contributions presented in this study are included in the article. Further inquiries can be directed to the corresponding author.

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
