# Peer review of "Increasing Nebulizer Spray Efficiency Using a Baffle with a Conical Surface: A Computational Fluid Dynamics Analysis"

_bioengineering, 2025, doi:10.3390/bioengineering12070680_

Round 1
Reviewer 1 Report
Comments and Suggestions for Authors
1. How did you validate the results of the CFD simulation?
2. For any CFD modeling, it is necessary to do a mesh independance test.
3. Are you sure that Ansys Fluent is FEM-based software? Not FVM?
4. The quality of the figures are very low and it is difficult to reconize the colors and regions, specifically Figures 7, 8, and 10.
5. You should expand the Conclusion section.
6. The results and discussion should more descriptive.
7. The initial and boundary conditions of the modeling are not clearly described.
8. Check the following papers: https://doi.org/10.1016/j.jaerosci.2025.106615; https://doi.org/10.1038/s41598-022-10369-8
9. Improve the English written of the maniscript.
Author Response
Comment 1.1: How did you validate the results of the CFD simulation?
Author Response: In this study, the performance of FEM model was validated by compared the results of physical spray distance and spray mass experiments. We added this explanation to the last sentence of page 5, Lines 142-143.
Comment 1.2: For any CFD modeling, it is necessary to do a mesh independence test.
Author Response: We thank the reviewer for this comment. According to a previous study (Med. Bio. Eng. Comput. 2006, 44, 785-792), the convergence criterion is met when changing the element size three consecutive times no longer affects the computational results. Such an element size setting meets the accuracy requirements stated in the literature (Materials 2023, 16, 2555). We added these statements on page 3, Lines 108-111. In addition, two new references were added to the revised manuscript.
Comment 1.3: Are you sure that Ansys Fluent is FEM-based software? Not FVM?
Author Response: We thank the reviewer for pointing this mistake out. The ANSYS FLUENT is an FVM-based software. We revised this mistake throughout the manuscript.
Comment 1.4: The quality of the figures are very low and it is difficult to reconize the colors and regions, specifically Figures 7, 8, and 10.
Author Response: Figures 7, 8, 9 and 10 were replaced with new pictures with hiher resolution.
Comment 1.5: You should expand the Conclusion section.
Author Response: In the revised manuscript, the conclusion section was expand as Line 309-317.
Comment 1.6: The results and discussion should more descriptive.
Author Response: In the revised manuscript, we added a new paragraph to discuss why the Con-0.25 baffle design demonstrates the best aerosol drug delivery performance has been added to the last paragraph on page 10 (Lines 286-297). In addition, the limitation of current study was added to Lines 303-307.
Comment 1.7: The initial and boundary conditions of the modeling are not clearly described.
Author Response: In this study's CFD simulation, the CFD model remains stationary. Therefore, the degrees of freedom in the X, Y, and Z directions for the nodes on the boundary wall surface were set to zero. We added this explanation to page 3, Lines 111-113.
Comment 1.8: Check the following papers: https://doi.org/10.1016/j.jaerosci.2025.106615; https://doi.org/10.1038/s41598-022-10369-8
Author Response: We thank the reviewer for this comment. The work of Rahimi‑Gorji et al. (Sci. Rep. 2022, 12, 6305) was added to the revised manuscript as Ref. 29.
Comment 1.9: Improve the English written of the maniscript.
Author Response: The English of the revised manuscript was checked by an English teacher whose mother language is English.
Reviewer 2 Report
Comments and Suggestions for Authors
Title: Increasing nebulizer spray efficiency using a baffle with a conical surface: A computational fluid dynamics analysis
Summary: Breath-actuated nebulizers used in aerosol therapy are vital to children and patients with disabilities and stand out for their ability to accurately deliver medication while minimizing waste. Their performance can be measured according to mass output and droplet size. This study aimed to analyze how baffle impact surface geometries affect pressure and flow streamlines inside the nebulizer using computational fluid dynamics (CFD). Computer-aided design models of conical symmetric, conical asymmetric, and arc-shaped baffle designs were analyzed using finite element CFD simulations, with optimal spray output validated through differences in mass.
Overall: The paper is easy to follow and has interesting results on different baffle shapes for nebulizer. Please ensure consistency with Figure Labels such as Figure 1 and referencing Figure 1 in the text as "Figure 1" are the same and not abbreviated.
The following comments are here to improve the paper.
Introduction
Line 49 - 77: Please rewrite the last 2 paragraphs to clearly state CFD being used on different nebulizer designs and then limitations of current design and then lead into your aim of this study. It already states limitations on breath nebulizers but does not have any discussion of CFD use on these type of nebulizers or other designs.
Methods
Line 97-99: Please add annotations on Fig. 1 with arrows to show the boundary walls, inlet flow port, drug spout port to follow the paper.
Line 109-116: Are there any boundary conditions applied to the walls of the nebulizer such as no slip conditions.
Line 120-121: It is unclear where the boundary walls, inlet and outlet ports are located on the device. Please label them in Fig. 2.
Line 147-148: Was there a Post Hoc test such as Tukey or Bonferonni to determine differences between groups? Is the flat baffet being used as a control design for comparison in Figure 12? Please add to paper to clarify in results.
Line 154: What is the horizontal distance between the vertical lines such as 1 mm spacing between lines? Please clarify in Figure 5 caption.
Line 185: How much dead volume is in this design and compared to other designs?
Line 266: Please clarify, how the Con-0.25 is it more efficient such as it is more efficient on mass difference?
Figure 7: The Pressure scale is difficult to read. Please make the legend clearer in the figure.
Figure 11: Correct the name Baffet to Baffle.
Figure 12: Correct the name Baffet to Baffle.
Author Response
Reviewer 2:
Comment 2.1: Overall: The paper is easy to follow and has interesting results on different baffle shapes for nebulizer. Please ensure consistency with Figure Labels such as Figure 1 and referencing Figure 1 in the text as "Figure 1" are the same and not abbreviated.
Author Response: We sincerely thank the reviewer for his comments. In the revised manuscript, “Fig. #” was replaced by “Figure #”.
Comment 2.2: Line 49 - 77: Please rewrite the last 2 paragraphs to clearly state CFD being used on different nebulizer designs and then limitations of current design and then lead into your aim of this study. It already states limitations on breath nebulizers but does not have any discussion of CFD use on these type of nebulizers or other designs.
Author Response: We sincerely thank the reviewer for his comments. The current status of the CFD study of the nebulizer was added to Lines 74-77. Several new references [27-30] were added. The limitation of CFD analysis on the nebulizer was added to Lines 303-307.
Comment 2.3: Line 97-99: Please add annotations on Fig. 1 with arrows to show the boundary walls, inlet flow port, drug spout port to follow the paper.
Author Response: The annotations with arrows of the outlets in the CFD model have been added to Figure 1.
Comment 2.4: Line 109-116: Are there any boundary conditions applied to the walls of the nebulizer such as no slip conditions.
Author Response: In this study's CFD simulation, the CFD model remains stationary. Therefore, the degrees of freedom in the X, Y, and Z directions for the nodes on the boundary wall surface were set to zero. We added this explanation to page 3, Lines 111-113.
Comment 2.5: Line 120-121: It is unclear where the boundary walls, inlet and outlet ports are located on the device. Please label them in Fig. 2.
Author Response: The annotations with arrows of the outlets in the CFD model have been added to Figure 1.
Comment 2.6: Line 147-148: Was there a Post Hoc test such as Tukey or Bonferonni to determine differences between groups? Is the flat baffet being used as a control design for comparison in Figure 12? Please add to paper to clarify in results.
Author Response: In this study, Scheffe post hoc test was performed to demonstrate the difference. This statement was added to page 5, Lines 155-157. For the experiment of Figure 12a, a flat baffle was used as a control design. We added an explanation: “A flat baffle was used in the nebulizer” to page 5, Lines 150 and the legend of Figure 12.
Comment 2.7: Line 154: What is the horizontal distance between the vertical lines such as 1 mm spacing between lines? Please clarify in Figure 5 caption.
Author Response: In Figure 5, the vertical line on the background provided reference markers. The distance between the red lines is 6 cm. We read the data by measuring the distance from the start and final points of the background markers. This explanation was added to the figure legend.
Comment 2.8: Line 185: How much dead volume is in this design and compared to other designs?
Author Response: The dead volume defined in the previous version manuscript is incorrect. Throughout the revised manuscript, we removed all the phrase “dead volume” and replaced it with “surface of liquid medication” . The change was made on page 7, Lines 196-211.
Comment 2.9: Line 266: Please clarify, how the Con-0.25 is it more efficient such as it is more efficient on mass difference?
Author Response: The reason why the Con-0.25 baffle design demonstrates the best aerosol drug delivery performance has been added to the last paragraph on pages 10-11 (Lines 286-297).
Comment 2.10: Figure 7: The Pressure scale is difficult to read. Please make the legend clearer in the figure.
Author Response: Figures 7-10 were replaced by a new picture with higher resolution.
Comment 2.11: Figure 11: Correct the name Baffet to Baffle. Figure 12: Correct the name Baffet to Baffle.
Author Response: We sincerely thank the reviewer for pointing out this typo error. “Baffet” was revised to “Baffle” throughout the manuscript.
Reviewer 3 Report
Comments and Suggestions for Authors
- How can computational fluid dynamics (CFD), specifically using the finite element method (FEM), be applied to improve the spray output efficiency of breath-actuated nebulizers, particularly for patients with low respiratory effort such as young children?
- How does the design of the baffle, particularly the Con-0.25 conical baffle, influence airflow patterns and pressure distribution inside a breath-actuated nebulizer, and what impact does this have on aerosol output efficiency?
- What role does dead volume play in reducing nebulizer efficiency, and how do different baffle geometries, such as conical or arc shapes, help minimize aerosol loss to these areas?
- What limitations in this CFD study could be addressed in future research to better evaluate nebulizer performance, especially in terms of aerosol droplet size and factors like solution temperature, viscosity, and gas velocity?
Author Response
Reviewer 3:
Comment 3.1: How can computational fluid dynamics (CFD), specifically using the finite element method (FEM), be applied to improve the spray output efficiency of breath-actuated nebulizers, particularly for patients with low respiratory effort such as young children?
Author Response: Breath-actuated nebulizers are generally not recommended for children under the age of five, as their breathing effort is often insufficient to activate the trigger valve. Although this study does not directly analyze the actuation of the trigger valve in breath-actuated nebulizers, the CFD-based analysis presented in this study may serve as a reference for structural designs aimed at reducing the activation force. This could help address the issue of breath-actuated nebulizers being unsuitable for younger children. These statements were added to page 11, Lines 309-315.
Comment 3.2: How does the design of the baffle, particularly the Con-0.25 conical baffle, influence airflow patterns and pressure distribution inside a breath-actuated nebulizer, and what impact does this have on aerosol output efficiency?
Author Response: In the revised manuscript, the mechanism of Con-0.25 conical baffle shows better aerosol output efficiency was added to pages 10-11, Lines 286-297.
Comment 3.3: What role does dead volume play in reducing nebulizer efficiency, and how do different baffle geometries, such as conical or arc shapes, help minimize aerosol loss to these areas?
Author Response: The dead volume defined in the previous version manuscript is incorrect. Throughout the revised manuscript, we removed all the phrase “dead volume” and replaced it with “surface of liquid medication” . The change was made on page 7, Lines 196-209.
Comment 3.4: What limitations in this CFD study could be addressed in future research to better evaluate nebulizer performance, especially in terms of aerosol droplet size and factors like solution temperature, viscosity, and gas velocity?
Author Response: To address this issue, the introduction of two-phase (solid–liquid mixture) CFD analysis will be a direction for future research. We added this statement to page 11, Lines 302-303.
Round 2
Reviewer 1 Report
Comments and Suggestions for Authors
Well-revised